# Associations between Sweet Taste Sensitivity and Polymorphisms (SNPs) in the *TAS1R2* and *TAS1R3* Genes, Gender, PROP Taster Status, and Density of Fungiform Papillae in a Genetically Homogeneous Sardinian Cohort

**DOI:** 10.3390/nu14224903

**Published:** 2022-11-19

**Authors:** Melania Melis, Mariano Mastinu, Lala Chaimae Naciri, Patrizia Muroni, Iole Tomassini Barbarossa

**Affiliations:** Department of Biomedical Sciences, University of Cagliari, 09042 Monserrato, CA, Italy

**Keywords:** sweet taste sensitivity, *TAS1R2* and *TAS1R3* genes, PROP phenotype and genotype, gender

## Abstract

Individual differences in sweet taste sensitivity can affect dietary preferences as well as nutritional status. Despite the lack of consensus, it is believed that sweet taste is impacted by genetic and environmental variables. Here we determined the effect of well-established factors influencing the general taste variability, such as gender and fungiform papillae density, specific genetic variants (SNPs of *TAS1R2* and *TAS1R3* receptors genes), and non-specific genetic factors (PROP phenotype and genotype), on the threshold and suprathreshold sweet taste sensitivity. Suprathreshold measurements showed that the sweet taste response increased in a dose-dependent manner, and this was related to PROP phenotype, gender, *rs35874116* SNP in the *TAS1R2* gene, and *rs307355* SNP in the *TAS1R3* gene. The threshold values and density of fungiform papillae exhibited a strong correlation, and both varied according to PROP phenotype. Our data confirm the role of PROP taste status in the sweet perception related to fungiform papilla density, show a higher sweet sensitivity in females who had lower BMI than males, and demonstrate for the first time the involvement of the *rs35874116* SNP of *TAS1R2* in the sweet taste sensitivity of normal weight subjects with body mass index (BMI) ranging from 20.2 to 24.8 kg/m^2^. These results may have an important impact on nutrition and health mostly in subjects with low taste ability for sweets and thus with high vulnerability to developing obesity or metabolic disease.

## 1. Introduction

Taste is the sensory modality considered to be one of the most relevant factors that influences nutrition and health [1,2,3]. This role is based on data showing that taste varies importantly among individuals influencing food preferences and therefore eating behavior. Many studies have focused on understanding and identifying the factors contributing to these large inter-individual differences [4,5,6,7]. It is widely known that age and gender can determine taste differences [8]. Some studies showed that women have a higher sensitivity than men [9,10,11] and had more taste fungiform papillae and buds [10,12]. On the other hand, the wide inter-individual variation in the density of fungiform papillae has been shown associated with differences in taste sensitivity [13,14,15,16,17,18,19,20]. Individual differences in taste function are also controlled by genetic factors. The best-known example of taste variability genetically controlled is the capacity to taste the bitterness of 6-n-propylthiouracil (PROP), which has been postulated as an oral marker of general taste perception, food predilections, and eating behavior, with subsequent effects on health [1,21,22,23,24,25,26,27,28,29,30,31,32,33,34,35,36,37,38]. It is well-established that subjects who experience PROP as extremely bitter (PROP super-tasters, ST) have a higher sensitivity to various tastes [21,22,23,24,28,39,40,41,42,43,44,45,46], more fungiform papillae [10,17,18,21,47,48], lower predisposition for several diseases [32,49,50,51], and exhibit particular longevity [34], compared to non-tasters (NT). Individual differences in the PROP taste sensitivity are primarily caused by the allelic variety of the PROP-binding bitter receptor gene, *TAS2R38* [4,52], which results in two major haplotypes: the dominant taster variant (PAV) and the recessive non-taster one (AVI) [4,27], as well as rare haplotypes with intermediate sensitivity [53,54].

On the other hand, the individual variations in sweet taste perception have been less studied, with controversial results. Sweet taste sensitivity has been associated with the genetic ability to perceive the prototypical taste stimuli, PROP [21,39,40,41]. However, other studies did not find this association [55,56,57]. Sweet taste has also been associated with sweet food intake [58], numbers of taste buds [13], age [59,60], hormonal status [61,62], and genetic variants of sweet receptors [63,64,65,66]. In particular, the sweet taste receptor is a heterodimer of two G-protein-coupled receptors, TAS1R2 and TAS1R3 [67,68,69,70,71], that are encoded by homologous genes located on chromosome 1 [72]. The *TAS1R2* gene shows a high allelic diversity [73]. However, few studies analyzed the effect of the allelic diversity of this gene on sweet taste sensitivity. Fushan and colleagues [64] analyzed 34 SNPs within *TAS1R2*, and none was associated with the threshold and suprathreshold sensitivity to sucrose [64]. Dias et al. [63] revealed that the *rs12033832* (G/A) of the *TAS1R2* gene affects sweet detection threshold, suprathreshold taste ratings, and sweet intake differently in subjects with obesity and normal weight subjects. However, other authors showed that the *rs12033832* SNP was indifferently associated with the sweet taste perception in obese and normal weight subjects [66]. BMI is used to define different weight groups in adults 20 years old or older. In normal weight subjects, BMI is 18.5 to 24.9, and in subjects with obesity, BMI is 30 or more. The missense SNP *rs35874116* (C/T), which results in the substitution of valine for isoleucine at codon 191 (Ile191Val) and is located in one of the putative binding sites of the protein [72,73,74,75], has been associated with sugar intake [76,77,78] but not with sucrose sensitivity [63,64]. Investigations on the effect of *TAS1R3* SNPs on sweet taste are mostly limited to its promoter region. Two SNPs located at positions −1572 C/T (*rs307355*) and −1266 C/T (*rs35744813*) upstream of the TAS1R3 coding sequence are in strong linkage disequilibrium, and they have been shown to correlate with taste sensitivity to sucrose accounting for 16% of the sweet taste variability [64]. Recently, other authors found an association between *rs35744813* SNP and suprathreshold sensitivity to sucrose [65], while no association with sucrose detection threshold was found [79]. In addition, a larger genome-wide association study could not replicate the above-described relationships between SNPs of both *TAS1R2* or *TAS1R3* and sweet taste sensitivity and/or sugar intake [80].

Considering these contentious claims, as well as the dietary benefits of sweets in the diet, it would be of great interest to shed light on factors that may influence sweet taste variability. Here we determined the effect of well-established factors influencing the general taste variability, such as gender and fungiform papillae density, specific genetic variants (SNPs of *TAS1R2* and *TAS1R3* receptors genes), and non-specific genetic factors (PROP phenotype and genotype), on sweet taste sensitivity.

To achieve this aim, sweet taste sensitivity was assessed by threshold and suprathreshold measurements which address detection and responsiveness at higher concentrations [7,81,82,83]. 

## 2. Materials and Methods

### 2.1. Subjects

In total, 107 Caucasian subjects (41 males, and 66 females, age 28.85 ± 4.04) were recruited through usual procedures at the University of Cagliari. They were originally from Sardinia, Italy. Subjects were normal weight with the body mass index (BMI) ranging from 20.2 to 24.8 kg/m^2^ and showed no variation in body weight larger than 5 kg over the previous 3 months. None of them were on a special diet, had food allergies, or took any medications that would have affected their ability to taste. To rule out any taste impairment, their taste function for the four basic tastes was tested using the taste strip test (Burghart Messtechnik, Wedel, Germany). All subjects were informed (verbally and in writing) regarding the aim and procedure of the study. They signed a consent form. The current study was carried out in accordance with the most recent version of the Helsinki Declaration, and all methods were authorized by the University Hospital Company’s (AOU) Ethical Committee in Cagliari, Italy.

### 2.2. Experimental Procedure

Each subject underwent tasting twice on two consecutive days; on the first day they were classified as PROP tasters, and a sample of the whole-unstimulated saliva (2 mL) was taken and preserved at −80 °C until molecular analyses were finished as described below. The density of fungiform papillae and sweet taste sensitivity were assessed on the second day. For at least 2 hours before testing, all subjects were required to refrain from eating, drinking anything save water, using oral care products, or chewing gum. To prepare for the test’s environmental conditions (23–24 °C; 40–50% relative humidity; light with standard solar light 15,000 lux), they had to be in the experiment room 15 min before the test’s scheduled start time (9.00 AM). The environmental conditions were kept constant throughout the experimental session. The testing environment was kept relatively quiet and odor-free during the examinations. Subjects were situated in comfortable chairs. To prevent alterations in taste function caused by the estrogenic phase, in women taste evaluations were performed around the sixth day of the menstrual cycle [84]. Stimuli were given as solutions in spring water, which were prepared 1–2 days before each session and kept in the refrigerator until 1 hour before testing and were administrated at room temperature. 

### 2.3. PROP Taster Status Classification

Two scaling methods were used to assign subjects to their respective PROP taster status. First, each subject underwent the three-solution test as described by Tepper et al. in 2001 [85], which has been supported by multiple studies [7,23,82,86,87]. The test consisted of the ratings of the perceived taste intensity to 3 suprathreshold sodium chloride (NaCl; 0.01, 0.1, 1.0 mol/L) (Sigma-Aldrich, Milan, Italy) and PROP (0.032, 0.32, and 3.2 mmol/L) (Sigma-Aldrich, Milan, Italy) solutions, which were collected by using the Labeled Magnitude Scale (LMS) [88]. The LMS allows subjects the option to compare the perceived strength of a taste stimulus to the strongest oral stimulus they have ever experienced. Concentration samples (10 mL) were offered in random order. Subjects were categorized as NT, MT, or ST depending on whether their assessments of PROP and NaCl differed from one another (subjects who gave PROP greater ratings than NaCl were categorized as ST while who gave NaCl greater ratings than PROP were categorized as NT) or overlapped (MT). After 1 h, subjects were classified as belonging to a PROP taster category employing the impregnated paper screening test [89,90]. Briefly, two paper disks were placed for 30 s on the tip of the tongue, one with sodium chloride, NaCl (1.0 mol/L), and the other with PROP solution (50 mmol/L). Subjects who evaluated the PROP higher than 67 mm in LMS were classified as ST, those who evaluated the PROP lower than 15 mm on the scale were classified as NT, and all others were categorized as MT [90]. For both methods, the interstimulus interval was set at 60 s. The study only included subjects who were classified in the same way by the two screening methods. 

Based on their taster group assignments, 18 subjects were classified as ST (16.82%), 68 were MT (63.55%) and 21 were NT (19.63%). 

### 2.4. Sweet Taste Sensitivity Assessments

#### 2.4.1. Sucrose Detection Threshold Measurements

Detection thresholds for sucrose were assessed using a 3-alternative forced-choice (3-AFC) procedure. Test solutions were in quarter-log steps (0.25 log) ranging from 9 × 10^−6^ to 0.5 mol/L of sucrose dissolved in spring water. A 10 mL taste solution or spring water was delivered into plastic cups labeled with 3-digit codes for blinding. During each trial, the subjects were presented with 3 cups in a random order, 2 of which contained spring water (control solutions) and 1 of which contained a given sucrose concentration. Subjects were instructed to swish the entire contents of one cup in their mouth for 5 s and then to spit it out. After tasting all 3 samples, they were asked to choose which one was different from the other 2 samples. The starting solution was 0.028 mol/L sucrose. The sucrose concentration presented was reduced after two correct responses and increased after a single incorrect response. The detection threshold was designated as the lowest concentration at which the subject correctly identified the target stimulus on three consecutive trials. The interstimulus interval as well as the intertrial interval was set at 60 s. Subjects were always asked to rinse their mouth with spring water before each test and between trials.

#### 2.4.2. Suprathreshold Sucrose Measurements

Five different suprathreshold sucrose solutions were used to evaluate the responsiveness to sucrose at higher concentrations. The suprathreshold sucrose solutions used were: 0.01 mol/L, 0.032 mol/L, 0.1 mol/L, 0.32 mol/L, and 1 mol/L. Solutions were presented for 30 s to subjects in a randomized order. Each subject was asked to rate the perceived intensity of each solution by using the general LMS (gLMS), which gives subjects the freedom to rate the intensity of a stimulus concerning any sensation. Subjects were asked to rinse their mouth between each solution. The interstimulus interval was set at 120 s. 

### 2.5. Density Assessments of the Fungiform Taste Papillae 

Fungiform papillae density was measured on the tip of the anterior tongue surface at the left side of the midline which provides measurements of the fungiform papilla density in high correlation with the total number on the tongue [48]. This method was the same one developed by Melis et al. 2013 [18] and is briefly described below. The tip of the tongue was dried with filter paper and stained by placing (for 3 s) a piece of filter paper (circle 6 mm in diameter) that was impregnated with the blue food dye (E133, Modecor Italiana, Italy). Photographic images of the stained area were taken using a Nikon Coolpix P520 (Centro Ufficio Service, Roma, Italy) (18.1 megapixels). The digital images were downloaded to a computer and analyzed using a “zoom” option in the Adobe Photoshop CS2 version 9.0 software (Adobe Systems Incorporated, San Jose, CA, USA). The fungiform papillae were identified by their mushroom shape and distinguished by their very light staining from filiform papillae which stained dark [13]. The number of papillae in the stained area was counted for each subject by three expert operators who were not informed of the genotype of SNPs analyzed and the PROP taster status of subjects [18,19,48]. Final density measurements were based on the consensus of the three observers which then calculated the density/cm^2^. 

### 2.6. Molecular Analysis

DNA was extracted from saliva samples using the standard salting-out procedure. The concentration of purified DNA was estimated by measuring the optical density at 260 nm with an Agilent Cary 60 UV-Vis Spectrophotometer (Agilent technologies Australia (M) Pty Ltd., Victoria, Australia). Subjects were genotyped for the *rs35874116* (C/T) which leads to the substitution of an isoleucine for valine at position 191 (Ile191Val) and *rs12033832* (A/G) of TAS1R2 gene and for the *rs307355* (C/T) and for the *rs35744813* (C/T) of *TAS1R3* gene which are located at position −1572 and −1266 upstream of the *TAS1R3* coding sequence. The *TAS2R38* gene’s *rs713598*, *rs1726866*, and *rs10246939*, which cause three amino acid substitutions (Pro49Ala, Ala262Val, and Val296Ile), were also genotyped in the subjects. These three loci result in two major haplotypes, PAV (the dominant taster variant) and AVI (the non-taster recessive one), as well as three uncommon haplotypes (AAI, AAV, and PVI). Molecular analyses were carried out by using a TaqMan^®^ SNP Genotyping Assay (Applied Biosystems by Life-Technologies Italia, Europe BV, Monza, Italy) according to the manufacturer’s specifications. The plates were read on a StepOne™ Real-Time PCR System following the manufacturer’s instructions (Applied Biosystems by Life Technologies Milano Italia, Europe BV, Monza, Italy). The results were analyzed by allelic discrimination of the sequence detector software (Genotyping—Applied Biosystems, version v2.3; by Life-Technologies Italia, Europe BV, Monza, Italy). Replicates, negative and positive controls were included in all reactions. The reactions were run on 96-well plates with fast thermal cycling conditions and the reagent concentrations were 1X TaqMan^®^ genotyping master mix (code: 4371355), 1X TaqMan^®^ genotyping assays (C_55646_20, C_25985586_10, C_25985586_10, C_3260593_10, C_8876467_10, C_9506827_10 and C_9506826_10 assay), 10 ng of DNA, and nuclease-free water.

### 2.7. Statistical Analyses

The genotype distribution and haplotype or allele frequency of the *TAS2R38*, *TAS1R2,* and *TAS1R3* SNPs were tested according to PROP taster status and according to gender by the Fisher method (Genepop software version 4.2; online software: http://genepop.curtin.edu.au/genepop_op3.html (accessed on 23 October 2022); Montpellier, France). A chi-squared goodness-of-fit test was used to establish whether the distribution of observed genotype frequencies agreed with those expected under the Hardy–Weinberg equilibrium. The Hardy–Weinberg expected frequency for each genotype of *TAS1R2* and *TAS1R3* was also calculated for p2, 2pq, and q2 by using Levene’s correction (GENEPOP program v. 4.2; online software: (http://genepop.curtin.edu.au/genepop_op5.html (accessed on 23 October 2022); Montpellier, France). This equation permits us to relate allele frequencies to genotype frequencies for the population.

Repeated-measures ANOVA was used to analyze differences in mean values ± SEM of the intensity ratings evoked by the five sucrose solutions. The same data were also analyzed according to PROP taster groups, gender, or SNPs of *TAS2R38*, *TAS1R2*, and *TAS1R3*. One-way ANOVA was used to analyze the effect of the *TAS2R38*, *TAS1R2*, and *TAS1R3* loci on the sucrose detection threshold according to groups. One-way MANOVA was used to compare differences in the fungiform papilla density and sucrose detection threshold related to PROP taster status. Post hoc comparisons were conducted with Fisher’s least significant difference (LSD) test unless the assumption of homogeneity of variance was violated, in which case Duncan’s test was used. 

The relationships between sweet taste sensitivity measured as sucrose threshold and as the perceived intensity ratings after stimulation with sucrose solution (1 mol/L) and density of fungiform papillae were assessed using Pearson linear correlation analysis.

Statistical analyses were conducted using STATISTICA for WINDOWS (version 7; StatSoft Inc., Tulsa, OK, USA). *p* values ≤ 0.05 were considered significant.

## 3. Results

Demographic and anthropometric features and the genotype and allele frequencies for the SNPs of *TAS2R38*, *TAS1R2,* and *TAS1R3* according to gender are shown in Table 1. One-way ANOVA showed that the BMI depends on gender (F_(1, 106)_ = 10.697, *p* = 0.00147), with males showing BMI values higher than females. No changes in age were found (*p* > 0.05). Molecular analysis at the *rs35874116* SNP of the *TAS1R2* gene identified 42 subjects being homozygous TT (191Val), 54 heterozygous CT, and 11 homozygous CC (Ile191). Molecular analysis at the *rs1203383* SNP identified 53 subjects being homozygous GG, 46 heterozygous GA, and 8 homozygous AA. Molecular analysis of the *rs307355* of *TAS1R3* gene SNP identified 94 subjects being homozygous CC and 13 heterozygous CT while no one carried the TT genotype. Molecular analysis for *rs35744813* of *TAS1R3* obtained only subjects homozygous for the major allele C. Molecular analysis at the three SNPs of the *TAS2R38* gene identified 22 subjects who had PAV/PAV genotype, 50 were heterozygous, and 26 had AVI/AVI genotype. A total of 7 participants with rare haplotypes were found (4 haplotypes AAV, 1 PVI, 1 AAI, and 1 AVV). Two subjects were not genotyped for this gene. No significant difference in the allele and genotype frequencies of *TAS2R38*, *TAS1R2,* and *TAS1R3* SNPs according to gender or according to PROP taster status was found (*χ^2^* < 5.60, *p* > 0.061, Fisher Exact test). 

The Hardy–Weinberg exact test showed that our cohort met the Hardy–Weinberg equilibrium for the three SNPs (*rs35874116 χ^2^* = 1.047, *p* = 0.592; *rs12033832 χ^2^* = 0.171 *p* = 0.918 and *rs307355 χ^2^* = 0.038, *p* = 0.982). In addition, Levene’s correction showed that the expected and observed frequencies for different *TAS2R38*, *TAS1R2,* and *TAS1R3* genotypes were overlapping, while it is not possible to calculate the Hardy–Weinberg equilibrium for the *rs35744813* SNP since only homozygous CC were found in the cohort.

The sweet intensity ratings evoked after stimulation with the five suprathreshold sucrose solutions are shown in Figure 1. Repeated measures of ANOVA revealed that the perceived intensity rating significantly increased as a function of the increase in concentration (F_(4, 428)_ = 315.71, *p* = 0.0000) (Figure 1). Post hoc comparison showed that the perceived intensity ratings for all concentrations were higher concerning those evoked by the preceding concentrations (*p* ≤ 0.00012; Fisher LDS). 

The effect of the concentration on the perceived intensity ratings after sucrose stimulation according to gender or PROP taster status is shown in Figure 2. Repeated-measures ANOVA revealed that the sweet intensity ratings of the five sucrose suprathreshold solutions varied by gender (F_(4, 424)_ = 77,949, *p* < 0.0001). Post hoc comparison showed that females gave higher perceived intensity ratings for the sucrose solutions 0.1, 0.32, and 1 mol/L than males (*p* ≤ 0.0076, Duncan’s test). The pairwise comparison also showed an increase of the perceived intensity ratings as a function of concentration, in both males and females, that was significant after the second concentration (0.032 mol/L) and continued to increase across all concentrations (p ≤ 0.038, Duncan’s test). In addition, post hoc comparison showed that the subjects classified as ST gave higher perceived intensity ratings to the sucrose solutions 0.032, 0.1, 0.32, and 1 mol/L than MT and NT (*p* ≤ 0.043, Fisher LDS, after repeated measures ANOVA), whose ratings did not vary from each other. Moreover, in ST subjects the intensity ratings significantly increased after the second concentration (0.032 mol/L) and continued to increase across all concentrations (*p* ≤ 0.0012, Fisher LDS, after repeated measures ANOVA), while in MT and NT the effect of concentration on the intensity ratings was significant after the third concentration (0.1 mol/L) (*p* ≤ 0.0056, Fisher LDS, after repeated measures ANOVA). No effect of *TAS2R38* SNPs on the perceived intensity ratings of the five sucrose suprathreshold solutions was found. 

The effect of the *rs35874116* and *rs12033832* SNPs of *TAS1R2* and *rs307355* SNP of *TAS1R3* on the perceived intensity ratings of the five sucrose suprathreshold solutions are shown in Figure 3. Specifically, subjects with genotype CC of the *rs35874116* SNP gave significantly higher intensity ratings to the sucrose solution 0.1 mol/L than subjects with genotype homozygote for the allele T (*p* = 0.027, Fisher LDS, after repeated measures ANOVA). Moreover, in the CC subjects, the sweet intensity rating increased after the second concentration (0.032 mol/L) and continued to increase across all concentrations (*p* ≤ 0.027, Fisher LDS, after repeated measures ANOVA), while in subjects heterozygous and homozygous for T allele the intensity ratings increased after the third solution (0.1 mol/L) and after the fourth solution (0.32 mol/L), respectively (*p* ≤ 0.0015, Fisher LDS, after repeated measures ANOVA). No differences related to the *rs12033832* SNP were found (*p* > 0.05): the sweet intensity ratings increased after the third concentration (0.1 mol/L) and continued to increase across all concentrations in the three genotypes (*p* ≤ 0.037, Fisher LDS, after repeated measures ANOVA). In subjects with genotype CC of the *rs307355* SNP, the intensity ratings increased after the third concentration (0.1 mol/L) and continued to increase across all concentrations (*p* ≤ 0.0015, Fisher LDS, after repeated measures ANOVA), while in heterozygous subjects the ratings increased after the fourth solution (0.32 mol/L) (*p* ≤ 0.00078 Fisher LDS, after repeated measures ANOVA). 

Figure 4 shows the scatterplots depicting the relationship between the density of fungiform papillae and sucrose threshold (A) and perceived intensity ratings for sucrose solution (1 mol/L) (B). The threshold values were linearly correlated with the density of fungiform papillae (*r* = 0.231; *p* = 0.0168) (Figure 4A), while no correlation was found between the suprathreshold intensity ratings and papilla density (*r* = 0.008; *p* = 0.932) (Figure 4B). 

The mean values ± SEM of the sucrose threshold (mmol/L) and density of fungiform papillae (No./cm^2^) in ST, MT, and NT are shown in Figure 5. One-way MANOVA showed that the sucrose threshold and density of fungiform papillae varied with PROP taster status (Wilks lambda = 0.88714, F_(4, 206)_ = 3.1778, *p* = 0.01465). Post hoc comparison showed that PROP NT had a lower density than ST and MT (*p* ≤ 0.014; Fisher LSD test) and that the sucrose threshold of ST was lower than that of NT (*p* = 0.021; Duncan’s test subsequent one-way ANOVA). 

No difference in sucrose threshold related to gender or the SNPs of *TAS2R38*, *TAS1R2,* and *TAS1R3* were found (*p* > 0.05; data not shown). 

## 4. Discussion

The first goal of this work was to study the effects on sweet taste sensitivity of well-known factors involved in taste variability, such as fungiform papillae density and gender, as well as that of specific and non-specific genetic factors in a genetically homogeneous Sardinian cohort. We used threshold and suprathreshold measures, which address detection and responsiveness at higher concentrations [7,81,82,83] and measure different characteristics of the gustatory system that may be important for understanding the relationships between genetic factors and taste sensitivity [81]. 

Our results showed that the perceived intensity ratings after stimulation with five suprathreshold sucrose solutions increased in a dose-dependent manner and this effect was associated with PROP phenotype, gender, *rs35874116* SNP in the *TAS1R2* gene, and *rs307355* SNP in *TAS1R3*. Specifically, the increase in the perceived sweetness intensity was mostly observed in STs. For all concentrations (except the lowest, 0.01 mol/L) STs’ ratings were higher than those of MT and NT. Furthermore, they exhibited intensity ratings that increased already after the second concentration (0.032 mol/L), while in MTs and NTs the perceived intensity increased only after the third concentration (0.1 mol/L). However, the dose-dependent effect of PROP phenotype on sucrose sweetness intensity was not due to *TAS2R38* genotypes. These findings confirm previous studies showing STs to exhibit a higher perception of sweets than NTs [21,39,40,41] and a correlation between the perceived intensity for sucrose and that for PROP, without associations with *TAS2R38* SNPs [91].

Differently from studies showing no differences in sweet taste perception between males and females [92,93,94,95], our results showed that the dose-dependent effect of gender was mostly observed in females, who give higher intensity ratings to the three higher concentrations of sucrose compared to males. It is interesting to note that females in our cohort have a lower BMI than males even though both were normal weight. An explanation of why males tend to have a higher BMI than females is provided by data showing lower sensitivity to sweets associated with increased hedonic response to high-sugar diets and higher energy consumption [96], mostly in males who prefer sweet foods more than females [97].

In addition, our results showed, for the first time, the involvement of the *rs35874116* of the *TAS1R2* gene in sweet taste variability. In fact, to date this SNP has been associated with sugar intake [76,77,78] but not with sweet sensitivity [63,64]. We found that homozygous TT subjects, who have Ile/Ile variant in the receptor, are not able to discriminate low sucrose concentrations, different from homozygous CC (with Val/Val variant), who showed an increase of perceived intensity as a function of concentrations already after 0.032 mol/L solution. Therefore, subjects with homozygous CC genotype were able to discriminate low sucrose also. Furthermore, the homozygous CC subjects gave intensity ratings higher than those of TT subjects at the concentration of 0.1 mol/L. This difference was not found at higher concentrations which evoked higher intensity ratings with respect to the previous concentration in other genotype groups also. The reason why the substitution of an isoleucine for valine in the receptor TAS1R2 should lead to a more facility to perceive low concentrations of sweet stimuli should be investigated. Some studies observed no association between this SNP and sweet taste [63,64], while recently it has been shown that this SNP reduces the levels of sweet taste receptors in the plasma membrane, leading to partial loss of function [98]. Due to this, normal weight subjects with the CC genotype showed reduced glucose excursions in response to the consumption of a glucose load [98]. Others found an association of this SNP with the intake of sugar [76,77,78]. In particular, overweight and obese subjects with the CC genotype, consumed less sugar [76,77] and exhibited a higher carbohydrate intake and had hypertriglyceridemia more than those of other genotypes [78]. Furthermore, diabetic subjects with CC genotype had lower HbA1c levels, which measures the development of glycemic burden and predicts complications of diabetes, indicating that this variant may confer beneficial effects in the regulation of daily glucose levels [99]. These considerations allowed us to speculate that a lower sensitivity to sweet solutions, as shown in subjects with the TT genotype, may lead individuals to choose foods rich in sugars to reach a sufficient level of sweetness. 

We did not find an association across genotypes of the *rs12033832* SNP of the *TAS1R2* gene in our sample of normal weight subjects. The intensity ratings increased significantly in all genotype groups as a function of concentrations after the third concentration. This result is consistent with previous studies [63,64]; however, a specific effect of this SNP on sweet taste and sugar intake was found by Dias et al. related to BMI [63]. In individuals with a BMI ≥ 25, those homozygous for the G allele gave lower suprathreshold intensity ratings and showed higher sugar intake than those homozygous for the A allele [63]. These divergent results could be due to differences in BMI, which is known to influence taste sensitivity [100,101,102,103,104]. 

It has been shown an effect of two SNPs located in the promoter region of the *TAS1R3* gene (*rs35744813* and *rs307355*) on sweet taste sensitivity [64]. Individuals who carry two T alleles in both SNPs displayed reduced sensitivity to sucrose compared to those who carry C alleles. This could be because the presence of the T allele results in reduced promoter activity with respect to the C allele and thus the ability to discriminate sucrose concentrations is diminished [64]. In our cohort, we observed a low frequency of heterozygous (12%) for *rs307355* SNP, no homozygous for the TT genotype, and 88% of homozygous for allele C. These differences in the frequency of genotype limit the possibility to find a significant effect of this SNP on sweet sensitivity. Nevertheless, we observed that in heterozygous individuals, the intensity ratings increased significantly as a function of concentrations after the fourth concentration, while in subjects with CC genotype the intensity ratings increased significantly already after the third concentration. However, we cannot rule out that wider differences may emerge in a cohort with a higher frequency of the T allele. We failed to replicate the association between sweet sensitivity and *rs35744813* SNP since all subjects of our sample carried the CC genotype. The C allele, which was associated with high taste sensitivity, represents the major allele in all geographical regions except Africa; in particular, the frequency of this SNP in the European population is CC 93% and TT 7%. So, it is not surprising that in our cohort, which comprises subjects from a genetically homogeneous sample from the island of Sardinia, the presence of the T allele is not present. 

According to previous studies [25,57], our results showed a linear and inverse correlation between sucrose threshold and fungiform papilla density. Contrary to expectations, we observed no effects of gender or *TAS2R38*, *TAS1R2,* and *TAS1R3* genes on the sweet threshold. Otherwise, a recent study showed a high sucrose detection threshold in obese subjects who carried the G allele in the *rs12033832* SNP of *TAS1R2* and a low detection threshold and intake of sugars in normal weight subjects who carried the G allele [63]. Our results showed significant differences in the sucrose threshold related to PROP taster status. STs exhibited significantly lower threshold values than NT subjects. This can be explained by the higher density of fungiform papillae that we found in ST than in NT, according to previous studies [10,17,18,21,47,48].

## 5. Conclusions

The present study is the first that analyzed together different genetic and non-genetic factors involved in sweet taste sensitivity in a genetically homogeneous population. Our findings confirm the role of PROP phenotype in the taste perception of sweets that could be due to differences in fungiform papillae among PROP taster groups and showed a higher sweet sensitivity of females who had lower BMI compared to males. In addition, our results showed the involvement of the *rs35874116* SNP of *TAS1R2* in the taste sensitivity to low concentrations of sweets in normal weight subjects. These results allowed us to speculate that males, PROP non-tasters, and subjects with TT genotype of the *rs35874116* SNP, having low sensitivity for sweets, could be more vulnerable to high sugar consumption and thus could have a great risk of increased BMI or metabolic disease. 

Recent findings suggested that sweet taste receptor TAS1R2/TAS1R3 is also expressed in many extra-oral tissues, including the intestine and pancreas, and plays important role in detecting nutrients and regulating metabolic processes involving the release of insulin [105]. Thus, studying the genetic variation of the sweet taste receptors genes may help us to better understand individual differences in predisposition to metabolic diseases, such as obesity and type 2 diabetes. Further studies are necessary to understand the impact of these SNPs on taste sensitivity and their relationship with metabolism.

## Figures and Tables

**Figure 1 nutrients-14-04903-f001:**
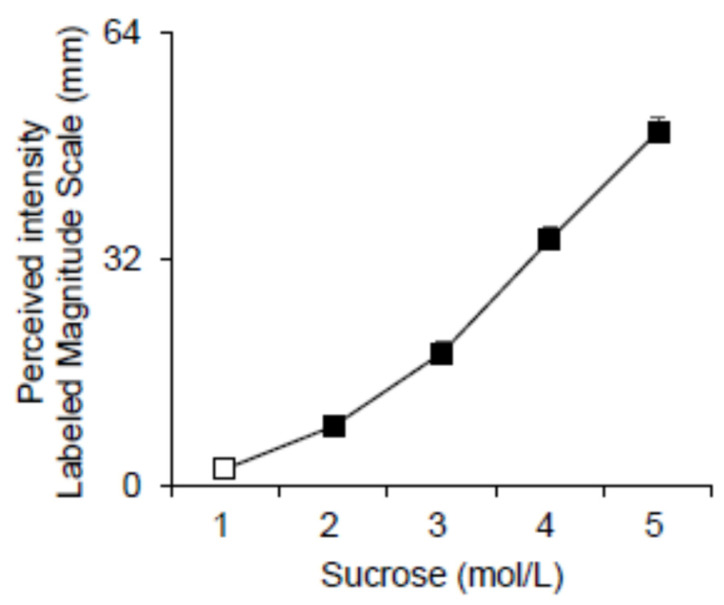
Mean values ± SEM of the perceived intensity ratings after stimulation with five suprathreshold sucrose solutions. The numbers 1, 2, 3, 4, 5 on the *X*-axis correspond to 0.01, 0.032, 0.1, 0.32, and 1 mol/L concentration, respectively. *n* = 107. The solid symbol indicates a significant difference concerning the previous value (*p* ≤ 0.00012; Fisher LDS, after repeated-measures ANOVA).

**Figure 2 nutrients-14-04903-f002:**
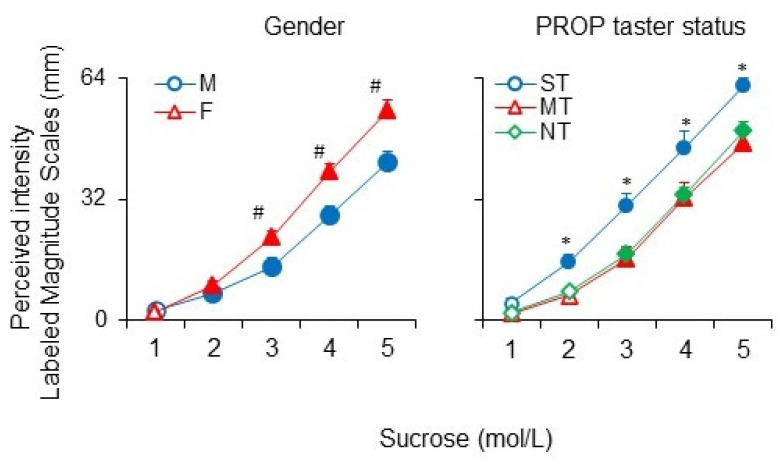
Mean values ± SEM of the perceived intensity ratings after stimulation with the five sucrose suprathreshold solutions according to gender and PROP Taster status. The numbers 1, 2, 3, 4, 5 on the *X*-axis correspond to 0.01, 0.032, 0.1, 0.32, and 1 mol/L concentration, respectively. *n* = 107. The solid symbol denotes a significant difference from the previous value (*p* ≤ 0.037; Fisher LDS, after repeated-measures ANOVA). # indicates a significant difference between the corresponding values of females and of males (*p* ≤ 0.0076; Duncan’s test, after repeated-measures ANOVA) * Indicate significant differences between the value of super tasters (ST) and the corresponding values of medium tasters (MT) and non-tasters (NT) (*p* ≤ 0.043; Fisher LDS, after repeated-measures ANOVA).

**Figure 3 nutrients-14-04903-f003:**
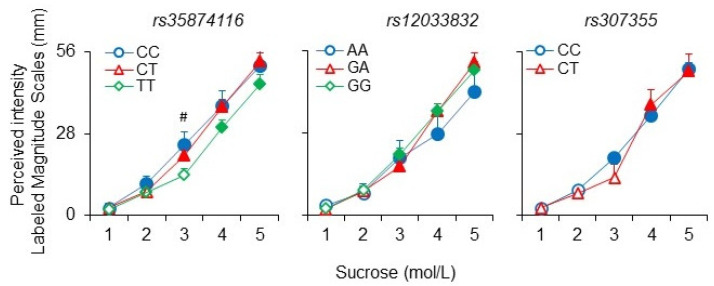
Mean values ± SEM of the perceived intensity ratings after stimulation with five suprathreshold sucrose solutions according to genotypes of *rs35874116* and *rs12033832* SNPs of TAS1R2 and *rs307355* SNP of TAS1R3. The numbers 1, 2, 3, 4, 5 on the *X*-axis correspond to 0.01, 0.032, 0.1, 0.32, and 1 mol/L concentration, respectively). *n* = 107. The solid symbol denotes a significant difference with respect to the previous value (*p* ≤ 0.037; Fisher LDS, after repeated-measures ANOVA). # indicates a significant difference between the value of CC subjects and the corresponding value of TT subjects (*p* = 0.028; Fisher LDS or Duncan’s test, after repeated-measures ANOVA).

**Figure 4 nutrients-14-04903-f004:**
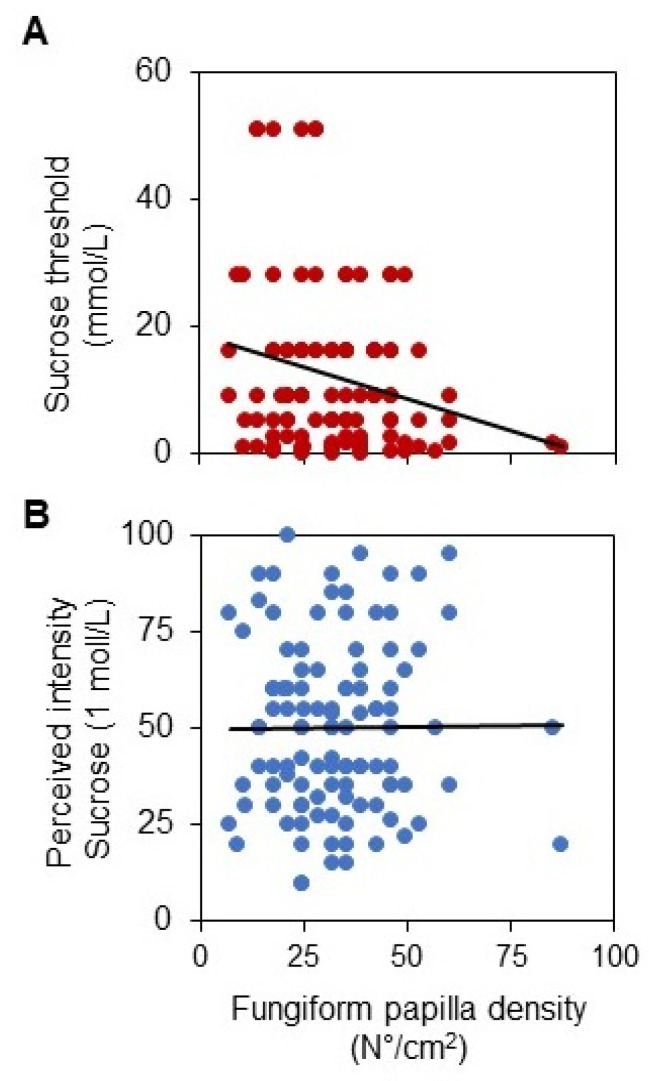
Linear correlation analysis between fungiform papillae density and the parameters describing sweet taste sensitivity. Relationship between the sucrose threshold and fungiform papillae density (*r* = 0.231; *p* = 0.0168, Pearson linear correlation) (**A**). Relationship between the perceived intensity ratings after stimulation with sucrose solution (1 mol/L) and density of fungiform papillae (*r* = 0.008; *p* = 0.932, Pearson linear correlation) (**B**).

**Figure 5 nutrients-14-04903-f005:**
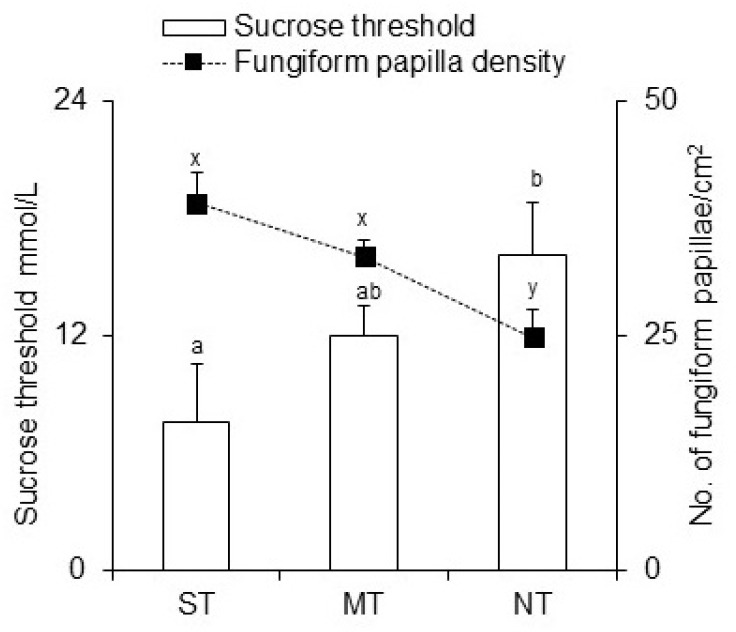
Sucrose threshold (mmol/L) and density of fungiform papillae (No./cm2) according to PROP taster status. All values are means (±SEM). *n* = 107. ST, super tasters; MT, medium tasters; and NT, non-tasters. Different letters indicate significant differences (a, b, and c are used to denote differences in sucrose threshold (*p* = 0.021, LSD test subsequent one-way MANOVA), x and y are used to denote differences in fungiform papillae density (*p* < 0.014 Duncan’s test, subsequent one-way MANOVA).

**Table 1 nutrients-14-04903-t001:** Demographic, anthropometric, and genetic features of subjects according to gender.

	Male (n = 41)	Female (n = 66)	*p*-Value
Age (y)	24.71 ± 0.66	24.28 ± 0.52	0.613
BMI (kg/m^2^)	22.95 ± 0.48 *	20.99± 0.37	0.0015
*TAS2R38 gene*			
Genotype n (%)			
PAV/PAV	10 (24.4)	12 (19.7)	0.510
PAV/AVI	18 (43.9)	32 (48.5)	
AVI/AVI	9 (22.0)	17 (27.3)	
Rare	4 (9.8)	3 (4.5)	
Haplotype n (%)			
PAV	38 (47.5)	58 (43.9)	0.489
AVI	38 (47.5)	71 (53.8)	
Rare	4 (5.0)	3 (2.3)	
*TAS1R2 gene*			
*rs35874116*			
Genotype n (%)			
TT	20 (48.8)	22 (33.3)	0.061
CT	19 (46.3)	35 (53.0)	
CC	2 (4.9)	9 (13.6)	
Allele n (%)			
T	59 (72.0)	79 (59.8)	0.086
C	23 (28.0)	53 (40.2)	
*rs12033832*			
Genotype n (%)			
GG	22 (53.7)	31(47.0)	1
GA	14 (34.1)	32 (48.5)	
AA	5 (12.2)	3 (4.5)	
Allele n (%)			
G	58 (70.7)	94 (71.2)	1
A	24 (29.3)	38 (28.8)	
*TAS1R3 gene*			
*rs307355*			
Genotype n (%)			
CC	36 (87.8)	58 (87.9)	1
CT	5 (12.2)	8 (12.1)	
TT	0	0	
Allele			
C	77 (93.9)	124 (93.9)	1
T	5 (6.1)	8 (6.1)	
*rs35744813*			
Genotype n (%)			
CC	41 (100)	66 (100)	-
CT	0	0	
TT	0	0	
Allele n (%)			
C	82 (100)	132 (100)	-
T	0	0	

Values are means ± SE or the number of subjects. *p*-value derived from one-way ANOVA or Fisher’s method. BMI: body mass index. *n* = 107. * Indicate significant differences between the value of males and females.

## Data Availability

The data presented in this study are available on request from the corresponding author. The data are not publicly available in accordance with consent provided by participants on the use of confidential data.

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
