# Peer review of "Associations between Sweet Taste Sensitivity and Polymorphisms (SNPs) in the TAS1R2 and TAS1R3 Genes, Gender, PROP Taster Status, and Density of Fungiform Papillae in a Genetically Homogeneous Sardinian Cohort"

_nutrients, 2022, doi:10.3390/nu14224903_

Round 1

Reviewer 1 Report

In the manuscript by Melis et al. the authors investigate the role of SNPs of TAS1R2 and 14 TAS1R3 receptors genes, PROP phenotype and genotype, gender and fungiform papillae density on sweet taste sensitivity.

The manuscript is in general of interest and well written.

However, I have some concerns, regarding this work:

-          In the count of TAS2R38 genotypes (lane 236, lane 242-244 and table 1) the total count of individuals is 104 instead of 107. Please check or explain the discrepancy.

-          Given the genetics peculiarity of Sardinian population, I would suggest to specify in the title of the manuscript, that this study is made in this specific genomic context. In general, this may also explain the discrepancies in genotypes distributions. Moreover, this will add major value to the study.

Author Response

Rebuttal to comments of Reviewer 1’ comments

We have reworked the manuscript according to Reviewer 1’ comments and suggestions.

In the revised manuscript the changes made according to Reviewer 1 are highlighted in red.

Reviewer #1

 Comments and Suggestions for Authors:

 In the manuscript by Melis et al. the authors investigate the role of SNPs of TAS1R2 and 14 TAS1R3 receptors genes, PROP phenotype and genotype, gender and fungiform papillae density on sweet taste sensitivity.

The manuscript is in general of interest and well written.

However, I have some concerns, regarding this work:

-          In the count of TAS2R38 genotypes (lane 236, lane 242-244 and table 1) the total count of individuals is 104 instead of 107. Please check or explain the discrepancy.

Reply: We apologize for the confusion and thank the reviewer who gave us the opportunity to make this correction. We corrected the number of subjects in the text (lines 240, 241 and 247) and added the following sentence at lines 249-250 : “Two subjects were not genotyped for this gene”.  The reason for this was that this polymorphism was analyzed as last when we had finished the DNA from two subjects

-          Given the genetics peculiarity of Sardinian population, I would suggest to specify in the title of the manuscript, that this study is made in this specific genomic context. In general, this may also explain the discrepancies in genotypes distributions. Moreover, this will add major value to the study.

Reply: We comply with the Reviewer’s request. We specified this in the title, and in the Discussion and Conclusions sections (lines 357-358 and 445).

Reviewer 2 Report

In this paper, authors studied the effect of well-established factors influencing the general taste variability, such as gender and fungiform papillae density, specific genetic variants (SNPs of TAS1R2 and TAS1R3 receptors genes) and non-specific genetic factors (PROP phenotype and genotype), on the threshold and suprathreshold sweet taste sensitivity. The introduction provides a relevant background for the understanding of the subject and the methods are well described with appropriate experimental design. Results are clearly described and discussion addresses all results and gives the main idea taken from the findings. For this reason I suggest small minor correction, such as these described below.

Page 1, lines 12-15. I suggest to rephrase the sentence as it is a bit confusing to understand. Maybe something like this: “Here we determined the effect of well-established factors influencing the general taste variability, such as gender and fungiform papillae density, specific genetic variants (SNPs of TAS1R2 and TAS1R3 receptors genes) and non-specific genetic factors (PROP phenotype and genotype), on the threshold and suprathreshold sweet taste sensitivity.”. The same goes for this bit in the introduction.

Maybe because I am not familiar with this, but what would a normal weight be? Would it be better maybe providing some weight range so the reader gets an idea of what would a normal weight be in practical terms?

Page 11, Lines 374-378.There is a typo in this sentence that says then instead of than. Additionally, it is clear what you meant, you say that an explanation of why males tend to have a high BMI than females must be provided, but you cite some papers. Please, clarify this.

Page 12, line 434: replace not genetic by non-genetic

Page 12, line 445: replace among which by including

Page 12, line 448: I suggest replacing “could play a role also in understanding” by “may help us to better understand”.

Author Response

Rebuttal to comments of Reviewer 2’ comments

We have reworked the manuscript according to Reviewer 2’ comments and suggestions.

In the revised manuscript the changes made according to Reviewer 2 are highlighted in blue.

Reviewer #2

 Comments and Suggestions for Authors:

In this paper, authors studied the effect of well-established factors influencing the general taste variability, such as gender and fungiform papillae density, specific genetic variants (SNPs of TAS1R2 and TAS1R3 receptors genes) and non-specific genetic factors (PROP phenotype and genotype), on the threshold and suprathreshold sweet taste sensitivity. The introduction provides a relevant background for the understanding of the subject and the methods are well described with appropriate experimental design. Results are clearly described and discussion addresses all results and gives the main idea taken from the findings. For this reason I suggest the publication of the manuscript, after small minor correction, such as these described below.

Page 1, lines 12-15. I suggest to rephrase the sentence as it is a bit confusing to understand. Maybe something like this: “Here we determined the effect of well-established factors influencing the general taste variability, such as gender and fungiform papillae density, specific genetic variants (SNPs of TAS1R2 and TAS1R3 receptors genes) and non-specific genetic factors (PROP phenotype and genotype), on the threshold and suprathreshold sweet taste sensitivity.”. The same goes for this bit in the introduction.

Reply: We comply with the Reviewer’s request. We rephrased accordingly (lines 13-16 and 84-87).

Maybe because I am not familiar with this, but what would a normal weight be? Would it be better maybe providing some weight range so the reader gets an idea of what would a normal weight be in practical terms?

Reply: We comply with the Reviewer’s request. We specified in the abstract the BMI of our normal-weight cohort (lines23-24), and in the Introduction (67-69) we added the following sentences: “BMI is used to define different weight groups in adults 20 years old or older. In nor-mal-weight subjects, BMI is 18.5 to 24.9, and in subjects with obesity, BMI is 30 or more”.

Page 11, Lines 374-378.There is a typo in this sentence that says then instead of than. Additionally, it is clear what you meant, you say that an explanation of why males tend to have a high BMI than females must be provided, but you cite some papers. Please, clarify this.

Reply: We comply with the Reviewer’s request. We corrected the type and rephrased as follows (lines 349- 380): “An explanation of why males tend to have a high BMI, than females, is provided by data showing …”.

Page 12, line 434: replace not genetic by non-genetic

Reply: We comply with the Reviewer’s request. We replaced not genetic” with “non-genetic” (line 444).

Page 12, line 445: replace among which by including

Reply: We comply with the Reviewer’s request. We replacedamong whichwithincluding (line 455).

Page 12, line 448: I suggest replacing “could play a role also in understanding” by “may help us to better understand”.

Reply: We comply with the Reviewer’s request. We replacedcould play a role also in understandingwithmay help us to better understand(line 458).

Reviewer 3 Report

This paper determined the effect on the threshold and suprathreshold sweet taste sensitivity of well-established factors influencing the general taste variability. The results showed that the sweet taste response increased in a dose-dependent manner, and it was related to PROP phenotype, gender, rs35874116 SNP in the TAS1R2 gene, and rs307355 SNP in the TAS1R3 gene. The threshold values and density of fungiform papillae exhibited a strong correlation. The work is meaningful and useful and might have an important impact on nutrition and health for human beings. However, there are still some concerns need to be addressed.

1. According to the results in Figure 3, authors conclude that the involvement of the rs35874116 SNP of TAS1R2 in the sweet taste sensitivity of normal weight subjects. However, this phenomenon was only seen when the concentration of the sucrose solution was 0.1 mol/L. Why subjects with genotype CC of the rs35874116 SNP did not give significantly higher intensity ratings to the other concentration of sucrose solution?

2. The graphic symbols presented are not consistent with those used in the figures (figure 1-3).

3. Why rs35874116 SNP in TAS1R2 plays an important role in sweet taste sensitivity? Was the protein sequence changed? Or was the gene expression changed?

Author Response

Rebuttal to comments of Reviewer 3’ comments

We have reworked the manuscript according to Reviewer 3’ comments and suggestions.

In the revised manuscript the changes made according to Reviewer 3 are highlighted in green.

Reviewer #3

 Comments and Suggestions for Authors:

This paper determined the effect on the threshold and suprathreshold sweet taste sensitivity of well-established factors influencing the general taste variability. The results showed that the sweet taste response increased in a dose-dependent manner, and it was related to PROP phenotype, gender, rs35874116 SNP in the TAS1R2 gene, and rs307355 SNP in the TAS1R3 gene. The threshold values and density of fungiform papillae exhibited a strong correlation. The work is meaningful and useful and might have an important impact on nutrition and health for human beings. However, there are still some concerns need to be addressed before publication.

  1. According to the results in Figure 3, authors conclude that the involvement of the rs35874116 SNP of TAS1R2 in the sweet taste sensitivity of normal weight subjects. However, this phenomenon was only seen when the concentration of the sucrose solution was 0.1 mol/L. Why subjects with genotype CC of the rs35874116 SNP did not give significantly higher intensity ratings to the other concentration of sucrose solution?

Reply: We recognize that we were not accurate with the discussion of this result which resulted in confusion.  The text We found that homozygous TT subjects, who have Ile/Ile variant in the receptor, are not able to discriminate low sucrose concentrations, differently from homozygous CC (with Val/Val variant), who showed an increase of perceived intensity as a function of concentrations already after 0.032 mol/L solution. Furthermore, the homozygous CC subjects gave intensity ratings higher than those of TT subjects at the concentration of 0.1 mol/L” has been changed (lines 386-394)  as follows: “We found that homozygous TT subjects, who have Ile/Ile variant in the receptor, are not able to discriminate low sucrose concentrations, differently from homozygous CC (with Val/Val variant), who showed an increase of perceived intensity as a function of concentrations already after 0.032 mol/L solution. Therefore, subjects with homozygous CC genotype were able to discriminate also low sucrose. Furthermore, the homozygous CC subjects gave intensity ratings higher than those of TT subjects at the concentration of 0.1 mol/L. This difference was not found at higher concentrations which evoked higher intensity ratings with respect to the previous concentration also in other genotype groups”.

  1. The graphic symbols presented are not consistent with those used in the figures (figure 1-3).

Reply: We apologize, but we don't understand what the reviewer means. Figures 2 and 3, which show data according to concentrations and gender and PROP status or polymorphisms, show different symbols with respect to those of figure 1 which shows data only according to concentration.

  1. Why rs35874116 SNP in TAS1R2 plays an important role in sweet taste sensitivity? Was the protein sequence changed? Or was the gene expression changed?

Reply: We point out to the reviewer that the effect of this polymorphism was specified in the introduction (line69-71), the missense SNP, rs35874116 (C/T) which results in the substitution of valine for iso-leucine at codon 191 (Ile191Val), and is located in one of the putative binding sites of the protein [72-75]) and in the methods section (lines 191-192) Subjects were genotyped for the rs35874116 (C/T) which leads to the substitution of an isoleucine for valine at position 191 (Ile191Val)).

The reason why the substitution of an isoleucine for valine in the receptor TAS1R2 should lead to a more facility to perceive low concentrations of sweet stimuli should be investigated. A sentence has been added in the discussion at lines 394-396.